# A Cluster of Paragonimiasis with Delayed Diagnosis Due to Difficulty Distinguishing Symptoms from Post-COVID-19 Respiratory Symptoms: A Report of Five Cases

**DOI:** 10.3390/medicina59010137

**Published:** 2023-01-10

**Authors:** Jun Sasaki, Masanobu Matsuoka, Takashi Kinoshita, Takayuki Horii, Shingo Tsuneyoshi, Daiki Murata, Reiko Takaki, Masaki Tominaga, Mio Tanaka, Haruhiko Maruyama, Tomotaka Kawayama, Tomoaki Hoshino

**Affiliations:** 1Department of Internal Medicine, Division of Respirology, Neurology, and Rheumatology, Kurume University School of Medicine, Kurume 830-0011, Japan; 2Department of Infectious Diseases, Division of Parasitology, Faculty of Medicine, University of Miyazaki, Miyazaki 889-1692, Japan

**Keywords:** pulmonary paragonimiasis, *Paragonimus westermani*, COVID-19, post-COVID-19 condition, delayed diagnosis, case cluster

## Abstract

Paragonimiasis caused by trematodes belonging to the genus *Paragonimus* is often accompanied by chronic respiratory symptoms such as cough, the accumulation of sputum, hemoptysis, and chest pain. Prolonged symptoms, including respiratory symptoms, after coronavirus disease 2019 infection (COVID-19) are collectively called post-COVID-19 conditions. Paragonimiasis and COVID-19 may cause similar respiratory symptoms. We encountered five cases of paragonimiasis in patients in Japan for whom diagnoses were delayed due to the initial characterization of the respiratory symptoms as a post-COVID-19 condition. The patients had consumed homemade drunken freshwater crabs together. One to three weeks after consuming the crabs, four of the five patients were diagnosed with probable COVID-19. The major symptoms reported included cough, dyspnea, and chest pain. The major imaging findings were pleural effusion, pneumothorax, and nodular lesions of the lung. All the patients were diagnosed with paragonimiasis based on a serum antibody test and peripheral blood eosinophilia (560–15,610 cells/μL) and were treated successfully with 75 mg/kg/day praziquantel for 3 days. Before diagnosing a post-COVID-19 condition, it is necessary to consider whether other diseases, including paragonimiasis, may explain the symptoms. Further, chest radiographic or blood tests should be performed in patients with persistent respiratory symptoms after being infected with COVID-19 to avoid overlooking the possibility of infection.

## 1. Introduction

Paragonimiasis is a food-borne parasitic disease caused by trematodes belonging to the genus *Paragonimus*, for which the majority are found in Asia. Humans can become infected by consuming raw or uncooked freshwater crabs, crayfish, or the meat of wild boars or deer. Although some cases are asymptomatic, the presenting symptoms typically include cough, the accumulation of sputum, and chest pain [1]. Delayed diagnosis has been reported due to the presence of atypical symptoms or an abnormal clinical course [2].

The coronavirus disease 2019 (COVID-19) pandemic has led to the delayed diagnosis of various diseases, including lung cancer [3], gynecological cancer [4], breast cancer [4,5], and gastrointestinal cancer [6]. COVID-19 can cause prolonged symptoms such as dyspnea and cough after infection, resulting in a post-COVID-19 condition [7]. As the number of post-acute COVID-19 patients has increased, so has the importance of excluding the possibility of other respiratory infections in individuals with persistent respiratory symptoms.

Herein, we report a cluster of five Japanese cases of paragonimiasis in which a diagnosis was delayed due to the mischaracterization of the symptoms as a post-COVID-19 condition.

## 2. Case Report

The clinical characteristics of the patients considered are summarized in Table 1. Of the patients, three were women and two were men, with a mean age of 39.2 years (range, 36–42 years). Cases 1 and 2 were housemates, and Cases 3 and 4 were a married couple. None of the patients had any relevant medical history, smoking history, or bronchial asthma. One patient (Case 4) had a history of allergic pollinosis.

All five patients consumed raw freshwater crabs (self-prepared drunken crabs) at a dinner party that they had attended together. One to three weeks after the party, four of the five patients presented to our hospital with complaints regarding a cough and were diagnosed with confirmed or likely COVID-19 (for two patients, this was confirmed by an antigen test). The mean time from the symptoms’ onset to the initial visit to our hospital was 5.4 months.

Three patients presented to their primary care physicians complaining of respiratory symptoms before visiting our hospital. All three patients were considered to be affected by post-COVID-19 symptoms. Of the three patients, two were placed under observation and one (Case 1), who had asthma-like symptoms including a fluctuating cough and dyspnea, was treated with inhaled corticosteroids (ICS)/long-acting β_2_ agonist (LABA) and 15 mg of oral prednisolone (PSL) for 3 weeks. The inhalation of ICS/LABA led to a temporary improvement in his symptoms.

At our hospital, the reported symptoms included cough (all patients), dyspnea (three patients), chest pain (two patients), fatigue (one patient), epigastric pain (one patient), and abdominal discomfort (one patient). Physical examination revealed a slight fever in Case 1 and bilateral decreases in breath sounds of the lower lung in Cases 1 and 4. Blood tests showed increased levels of eosinophils ranging from 560 to 15,610/μL (9.8–71.5%) in all five patients and increased levels of immunoglobulin E (IgE) ranging from 1857 to 3487 IU/L in three of the five patients. In contrast, C-reactive protein levels were near normal. Chest radiography and computed tomography (CT) showed pleural effusion (all patients), pneumothorax (three patients), nodular lesions with a cavity (one patient) and without cavity (one patient), infiltration of the lung (one patient), ascites (one patient), and a linear low attenuation area of the liver (one patient) (Figure 1A–E).

Pleural effusion fluid was examined in two patients. Both patients had exudative effusions, with a predominance of eosinophils and lymphocytes in Case 4 and lymphocytes in Case 5, examined by manual cell counting. A bronchoscopy was performed only in Case 1. To distinguish lung cancer and tuberculosis, a transbronchial lung biopsy of a right-sided nodule was performed. The biopsy specimen showed inflammatory changes, mainly eosinophils, without parasite eggs. Based on the finding of eosinophilia and the dietary history of raw freshwater crab consumption, we suggested that the symptoms were probably caused by a parasitic infection. We searched for parasite eggs in the pleural effusion and bronchial lavage fluid as well; however, we could not find any eggs. All five patients had high titers of serum *Paragonimus* IgG antibody, which was measured using a microplate enzyme-linked immunosorbent assay (ELISA). Cross-competition ELISA between *P. westermani* and *P. s. miyazakii* antigens suggested *P. westermani* infection.

All patients were treated with 75 mg/kg/day of praziquantel (PZQ) for 3 days, which led to an improvement in their symptoms and signs, including cough and dyspnea, eosinophilia, and pleural effusion. In Case 2, pleural fluid drainage was performed prior to PZQ administration.

## 3. Discussion

The incidence of paragonimiasis has decreased in Japan, and the number of cases diagnosed annually ranges from 17 to 49 [8]. Clustered or familial infections can occur [8,9]. According to a Japanese retrospective case review that considered 443 patients with paragonimiasis, the reported frequencies of symptoms were as follows: cough, 28.9%; sputum (including hemoptysis), 27.3%; chest pain, 18.5%; fever, 11.7%; dyspnea, 10.4%; and asymptomatic, 17.2% [1]. The most frequently observed finding on chest radiographs was pleural effusion (47.0%), followed by pneumothorax (16.9%), nodular shadows (11.5%), infiltrative shadows (8.8%), and mass shadows (6.5%) [1]. In the cases considered in this report, the symptoms and radiographic findings were consistent with those previously described. Furthermore, Case 5 appeared to have ectopic paragonimiasis in the liver.

A certain proportion of patients with COVID-19 experience sequelae after infection, which is known as long COVID [10] or post-acute COVID-19 syndrome [11]. The World Health Organization (WHO) has proposed that COVID-19-associated sequelae be referred to as a “post-COVID-19 condition” [7]. The prevalence of post-COVID-19 symptoms varies depending on the geographical area, survey method, and patient background. A Japanese cross-sectional study found the prevalence of cough, dyspnea, and chest pain to be 14.2%, 10.2%, and 2.4%, respectively (at a median duration of 29 days after the onset of COVID-19) [12]. The predominant CT pattern of COVID-19 revealed ground-glass opacity in the early stage and consolidation in later disease [13]. Pleural effusion and pneumothorax are uncommon in COVID-19 [14], a finding that differs from that of paragonimiasis. These findings were present in our patients.

In the cases considered in this study, the patients obtained freshwater crabs (the Japanese mitten crab) from a market. A meal of homemade drunken crabs was consumed by seven people. Six of the seven visited our hospital, with five suffering from paragonimiasis. Among the five patients with paragonimiasis, four had been previously diagnosed with or had been considered likely to be affected by COVID-19. Clearly, it was challenging for both patients and physicians to look past a diagnosis of a “post-COVID-19 condition”. Atypical findings of eosinophilia, pleural effusion, and pneumothorax helped to exclude the possibility that the patients were afflicted by a post-COVID-19 condition, leading to the correct diagnosis of paragonimiasis. These cases highlight the importance of educating citizens regarding the safe preparation of freshwater crab. In addition, health centers should alert markets that sell infected freshwater crabs to any potential danger. Moreover, physicians should be aware they may encounter patients with paragonimiasis due to the globalization of dietary cultures.

In Case 1, the asthma-like symptoms of cough and dyspnea improved immediately after the inhalation of ICS/LABA and oral PSL, although the patient had no history of bronchial asthma or allergy. The effectiveness of ICS/LABA and PSL may contribute to the delayed diagnosis of paragonimiasis. Harada et al. [15] reported a case of bronchodilator reversibility in the acute phase of paragonimiasis. Jeon et al. [16] described bronchoscopic findings in 13 patients with paragonimiasis with intrapulmonary parenchymal lesions; in total, seven patients showed bronchial luminal narrowing and congested or edematous mucosal changes. Among the seven patients, bronchial mucosal biopsies revealed chronic inflammation with eosinophilic infiltrations in three patients. These findings are similar to those observed in patients with bronchial asthma. Dyspnea in patients with paragonimiasis might be caused by an asthma-like mechanism and not only by pleural effusion or pneumothorax.

## 4. Conclusions

In conclusion, physicians should consider whether diagnoses other than post-COVID-19 conditions such as paragonimiasis may explain respiratory symptoms in patients suspected to have recently been infected with COVID-19. Physicians should examine chest radiographs and blood tests in patients with persistent respiratory symptoms after COVID-19 to aid correct diagnoses.

## Figures and Tables

**Figure 1 medicina-59-00137-f001:**
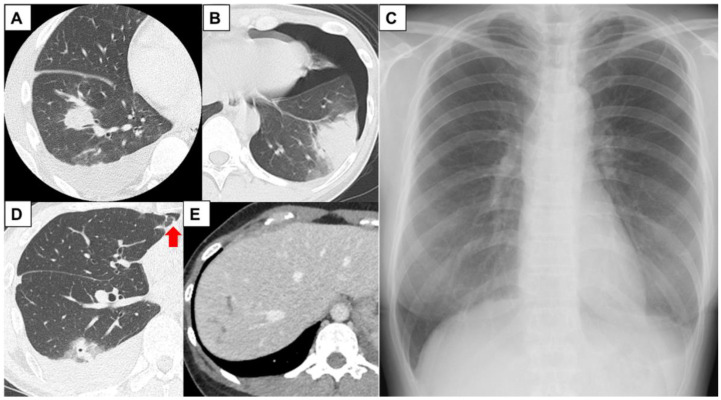
Imaging findings of cases. (**A**) Chest computed tomography (CT) of Case 1 showing a right-sided lung nodule and pleural effusion at the patient’s initial visit; (**B**) chest CT of Case 2 showing left-sided infiltration of the lower lung and pneumothorax 2 months prior to visiting our hospital; (**C**) chest X-ray of Case 3 showing left pleural effusion and pneumothorax; (**D**) chest CT of Case 4 showing a right-sided lung nodule with cavity, pleural effusion, and pneumothorax (arrow); (**E**) contrast-enhanced abdominal CT of Case 5 showing a linear low attenuation area of the liver, likely representing a pathway for worm migration.

**Table 1 medicina-59-00137-t001:** Clinical findings of the five cases considered.

	Case 1	Case 2	Case 3	Case 4	Case 5
Age (y), Sex	38, M	38, F	42, F	42, M	36, F
Symptoms					
Respiratory	Cough Dyspnea	Cough Dyspnea	Cough	Cough	Cough
Chest pain	+	+	−	−	−
Others	Fatigue	Epigastric pain	−	−	Abdominal discomfort
Radiographic and CT findings					
Intrapulmonary lesion	Nodular lesion	Infiltration	−	Nodular lesion	−
Pleural lesion	Pleural effusion	Pleural effusion Pneumothorax	Pleural effusion Pneumothorax	Pleural effusion Pneumothorax	Pleural effusion
Intraperitoneal lesion	Ascites	−	−	−	Linear low attenuation area of the liver
Peripheral blood					
WBC (cells/μL)	10,300	10,600	11,000	21,800	5700
Eosinophils (/μL)	5310	1290	5430	15,610	560
Eosinophil %	51.4%	12.2%	49.4%	71.5%	9.8%
IgE (IU/L)	1857	3487	96	2529	31
COVID-19 before diagnosis	Yes	Probably *	Yes	Probably *	No
Time from symptoms’ onset to diagnosis (mos)	5	5	5	6	6
Treatment					
PZQ	+	+	+	+	+
Others	−	Drainage of pleural fluid	−	−	−

COVID-19, coronavirus disease 2019; CT, computed tomography; IgE, immunoglobulin E; PZQ, praziquantel; WBC, white blood cell count. * Patients had been in close contact with persons with COVID-19 without a confirmatory test.

## Data Availability

Data supporting the study findings are available from the corresponding author on reasonable request.

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
