# Peer review of "A Cluster of Paragonimiasis with Delayed Diagnosis Due to Difficulty Distinguishing Symptoms from Post-COVID-19 Respiratory Symptoms: A Report of Five Cases"

_medicina, 2023, doi:10.3390/medicina59010137_

Round 1
Reviewer 1 Report
Dear Authors, Thank you for the manuscript. It is an interesting angle to highlight an importance of considering other possible diagnoses, in patients who presented with "long covid" symptoms in previously affected patients. It is also equally interesting to revisit this tropical infection that particularly acquired through ingestion of certain food type.
My feedback are as the followings:
1. For Figure 1B -I can't really appreciate pleural effusion, instead I noted a pneumothorax instead, is this correctly labelled? Similarly, for figure 1D-i can;t ttruly appreciate presence of pneumothorax. If an arrow/label can indicate where is it located within the figure that will be helpful.
2. For the 2 patients with COVID-19-how was the diagnosis confirmed? For the two other cases who are probable, why is it so?
Thank you
Reviewer 2 Report
I think that the article is well set up and scientifically valid. I have only a few minor comments.
Minor questions:
· In the results section describing blood tests (page 3, lines 72 to 75) the authors highlight the increase in eosinophils and IgE levels, typical of parasitic infestations and allergies. I would like to know if the cationic protein of eosinophils (ECP) has also been tested and if it would be suggested to add its values.
· In lines 88 to 90 (page 3) the authors describe the presence on pleural effusion of lymphocytes and eosinophils, in patient 4, and lymphocytes in patient 5. In my opinion, the authors should specify by what method the pleural effusion fluid was examined (e.g. if in flow cytometry). And if it was examined in cytometry, doing an immunophenotype, I would ask the authors to specify which types of lymphocytes were predominant (CD4 ?or CD8 ?) CD4 memory etc) if this data is available of course.
· Also in lines 91 to 97 (page 3) the authors emphasize the presence of IgG anti Paragonimus, and in biopsies the lack of eggs of the parasite. I would like to know if the parasite's eggs were also searched for in other biological samples such as sputum, feces or pleural fluid and whether they were found or not. It should be specified.
